

# Antibiotics alter development and gene expression in the model cnidarian *Nematostella vectensis*

Quinton Krueger[1,2], Britney Phippen[1] and Adam Reitzel[1,2]

[1] Biological Sciences, University of North Carolina at Charlotte, Charlotte, NC, United States of America
[2] Computational Intelligence to Predict Health and Environmental Risks (CIPHER) Center, University of North Carolina at Charlotte, Charlotte, NC, United States of America

## ABSTRACT

**Background**. Antibiotics are commonly used for controlling microbial growth in diseased organisms. However, antibiotic treatments during early developmental stages can have negative impacts on development and physiology that could offset the positive effects of reducing or eliminating pathogens. Similarly, antibiotics can shift the microbial community due to differential effectiveness on resistant and susceptible bacteria. Though antibiotic application does not typically result in mortality of marine invertebrates, little is known about the developmental and transcriptional effects. These sublethal effects could reduce the fitness of the host organism and lead to negative changes after removal of the antibiotics. Here, we quantify the impact of antibiotic treatment on development, gene expression, and the culturable bacterial community of a model cnidarian, *Nematostella vectensis*.

**Methods**. Ampicillin, streptomycin, rifampicin, and neomycin were compared individually at two concentrations, 50 and 200 $\mu$g mL$^{-1}$, and in combination at 50 $\mu$g mL$^{-1}$ each, to assess their impact on *N. vectensis*. First, we determined the impact antibiotics have on larval development. Next Amplicon 16S rDNA gene sequencing was used to compare the culturable bacteria that persist after antibiotic treatment to determine how these treatments may differentially select against the native microbiome. Lastly, we determined how acute (3-day) and chronic (8-day) antibiotic treatments impact gene expression of adult anemones.

**Results**. Under most exposures, the time of larval settlement extended as the concentration of antibiotics increased and had the longest delay of 3 days in the combination treatment. Culturable bacteria persisted through a majority of exposures where we identified 359 amplicon sequence variants (ASVs). The largest proportion of bacteria belonged to Gammaproteobacteria, and the most common ASVs were identified as *Microbacterium* and *Vibrio*. The acute antibiotic exposure resulted in differential expression of genes related to epigenetic mechanisms and neural processes, while constant application resulted in upregulation of chaperones and downregulation of mitochondrial genes when compared to controls. Gene Ontology analyses identified overall depletion of terms related to development and metabolism in both antibiotic treatments.

**Discussion**. Antibiotics resulted in a significant increase to settlement time of *N. vectensis* larvae. Culturable bacterial species after antibiotic treatments were taxonomically diverse. Additionally, the transcriptional effects of antibiotics, and after their removal result in significant differences in gene expression that may impact the physiology

Corresponding author
Quinton Krueger, qkrueger@uncc.edu

of the anemone, which may include removal of bacterial signaling on anemone gene expression. Our research suggests that impacts of antibiotics beyond the reduction of bacteria may be important to consider when they are applied to aquatic invertebrates including reef building corals.

# INTRODUCTION

Animals and bacteria interact in numerous ways depending on the specific bacteria (*Chan et al., 2019*; *Diaz & Restif, 2014*; *Vega & Gore, 2017*), host (*Berg, Zhou & Shapira, 2016*), and environment (*Dirksen et al., 2016*; *Samuel et al., 2016*). Bacterial communities associated with animals are typically diverse (*Dirksen et al., 2016*; *Huggett & Apprill, 2019*), remain distinct in composition from the surrounding environment (*Carrier & Reitzel, 2019*; *Marchioro et al., 2020*; *Woodhams et al., 2020*), and are to some extent host specific (*Ahern et al., 2021*; *O'Brien et al., 2020*). Additionally, microbiomes of host animals vary between geographically distinct populations, which may also be dependent on environmental conditions (*Faddetta et al., 2020*; *Ketchum et al., 2020*; *Williams et al., 2022*). The microbiome can also influence the hosts' physiological tolerance to environmental variation (*Fontaine & Kohl, 2023*; *Gilbert, Bosch & Ledón-Rettig, 2015*). This indicates that the life history of an individual, and its microbiome, are important factors for survival in extreme environments. However, most bacteria have not been characterized with respect to how they influence the life history of a host. The importance of determining the impact of specific bacteria is critical not only for identifying the beneficial bacteria that may facilitate resilience but also for controlling pathogenic bacteria that can be lethal to the host.

In marine habitats, antibiotics are applied to reduce or remove potential pathogenic bacteria from organisms of interest, such as fish (*Cabello, 2006*; *Salgado-Caxito et al., 2022*; *Yukgehnaish et al., 2020*), oysters (*Baralla et al., 2021*; *Salgueiro et al., 2021*), and corals (*Aeby et al., 2019*; *Hartman, Blackall & van Oppen, 2022*; *Neely et al., 2020*). Some groups of animals, including ecosystem engineers like corals, are particularly susceptible to rapid environmental changes and may become susceptible to opportunistic pathogens (*Gardner et al., 2019*). White Band Disease (WBD) is prominent in coral populations, which may be associated with pathogenic bacterium (*Gignoux-Wolfsohn & Vollmer, 2015*; *Sweet, Croquer & Bythell, 2014*). The applications of antibiotics arrested the progression of WBD, but necrotic tissues did not recover (*Sweet, Croquer & Bythell, 2014*). This result indicates that antibiotic application alone may not be adequate to save diseased individuals. Antibiotic treatment of the coral *Euphyllia paraivisa* resulted in differentially expressed genes related to development, cell communication and cell signaling, and persistent bacteria include members from the Alphaproteobacteria and Gammaproteobacteria classes (*Meron et al., 2020*). In *Pocillopora*, the reduction of the microbiome with antibiotic application in combination with heat stress resulted in elevated gene expression to heat response,

and increased modulation of the immune response (*Connelly et al., 2022*). Additionally, Rhodobacteraceae and Alteromonadaceae were the most prevalent bacteria that persisted through antibiotic treatment (*Connelly et al., 2022*). Further assessment of other host impacts such as development, persistent associated bacteria, and transcriptional responses will be beneficial to understand sublethal effects of antibiotics on cnidarians and other marine invertebrates.

Antibiotic use in aquaculture is projected to grow by more than 30% by 2030 to increase food production around the globe (*Schar et al., 2020*). However, the non-specific effects of antibiotics are poorly characterized in aquatic animals (*Yang, Song & Lim, 2020*). Antibiotics are commonly used in the field of gnotobiology, where germ-free individuals are studied to determine the roles of inoculated bacteria or microbiome (*Nass & Hamza, 2007*; *Provasoli & Shiraishi, 1959*; *Tinh et al., 2006*; *Xiang et al., 2013*). Studies of direct toxicity from antibiotics suggest that negative effects are more common for plants and bacteria than for animals (*Connelly et al., 2022*; *Lanzky & Halting-Sørensen, 1997*; *Nass & Hamza, 2007*). However, exposure of animals to antibiotics can also result in negative impacts on development and can impact molecular processes that are likely independent of the targeted bacteria (*Kohanski, Dwyer & Collins, 2010*). For example, aminoglycoside antibiotics treat infections caused by gram-negative bacteria but also result in ototoxicity and nephrotoxicity by damaging hair cells during development in vertebrates (*Huth, Ricci & Cheng, 2011*; *Rizzi & Hirose, 2007*). Exposure of *Branchiostoma belcheri*, a cephalochordate, to antibiotics resulted in sublethal effects including immunosuppression and a reduction in eicosanoids (*Yuan et al., 2015*). Moreover, antibiotics can directly affect the function and efficacy of mitochondria within the host cells (*Kovalakova et al., 2020*; *Miller & Singer, 2022*; *Singh, Sripada & Singh, 2014*). While antibiotics can reduce and potentially kill bacteria in a host, the direct effects of these compounds on the host are understudied (*Baralla et al., 2021*; *Bojarski, Kot & Witeska, 2020*; *Yang, Song & Lim, 2020*). Investigating the effects of antibiotics on bacterial survival and the host is necessary to understand the impact on the holobiont.

Here, we use three complementary approaches to measure the impact of antibiotic exposure on the cnidarian *Nematostella vectensis*. *N. vectensis* is a model species of sea anemone that has been used to experimentally determine the composition, dynamics, and potential functions of cnidarian bacterial communities (*Artamonova & Mushegian, 2013*; *Fraune, Forêt & Reitzel, 2016*; *Har et al., 2015*; *Mortzfeld et al., 2016*). *N. vectensis* has been insightful to determine how environmental variation, natural or chemical, can impact physiology and gene expression in sea anemones (*Berger et al., 2022*; *Elran et al., 2014*; *Reitzel et al., 2013*; *Reitzel et al., 2008a*; *Reitzel et al., 2008b*). The microbiome, which differs from the surrounding environment, is vertically transmitted (*Baldassarre et al., 2021*), changes throughout the life history stages of *N. vectensis* (*Mortzfeld et al., 2016*), and is impacted by anthropogenic chemicals including potassium nitrate and dioctyl phthalate (*Klein et al., 2021*). When enriched bacteria were added to axenic *N. vectensis* or concurrent with the native microbiome, all enriched species were lost after seven days (*Domin et al., 2018*), but transferred bacteria can confer higher temperature tolerance (*Baldassarre et al., 2022*). Other than microbiome manipulation, additional effects of one class of antibiotics,

aminoglycosides, have been identified to be toxic through the deterioration of hair bundles after exposure *N. vectensis* (*Menard, 2018*). The direct impacts of the antibiotic exposure on *N. vectensis* gene expression and the culturable microbiome following exposure have yet to be investigated. In this study, we determined the impact of different antibiotics and their combination on embryonic development and metamorphosis to the juvenile stage. Second, we measured and identified culturable bacteria using 16S rDNA gene sequencing after application of four individual antibiotics at two concentrations and in combination. Lastly, we measured the impact of an acute and chronic exposure to the antibiotic cocktail on the gene expression of adult *N. vectensis*. Together, this research reveals a complex developmental and transcriptional response to antibiotics in a cnidarian model that are important for interpreting future research on the interactions of cnidarians and bacteria.

## MATERIALS & METHODS

### Spawning of *Nematostella vectensis*

The animals used in this experiment were from a lab acclimated population of *Nematostella vectensis* derived from the original population collected from Rhode River, Maryland. Adult anemones were maintained in glass dishware in 15 parts per thousand (ppt) Artificial Seawater (ASW) (Instant Ocean) and fed freshly hatched *Artemia salina* three times per week (*Fritzenwanker & Technau, 2002*; *Hand & Uhlinger, 1992*; *Stefanik, Friedman & Finnerty, 2013*). Anemones were cultured at room temperature prior to experimentation. Culturing bowls were cleaned weekly and fresh artificial seawater was replaced at the same time.

For experiments using developmental stages, adult *N. vectensis* were spawned following a standard weekly protocol to procure embryos. Anemones were cultured for two days at 16 °C in an incubator without light. Anemones were then moved to room temperature, fed cubes of dissected mussel gonads or mantle, and then incubated at 25 °C overnight in an incubator under constant light. Gametes were shed the following morning and fertilization occurred in the bowl. Fertilized egg masses were moved to clean artificial seawater and maintained at 20 °C in the dark for 24–36 h. Adults were maintained under standard laboratory conditions for three days before returning to the 16 °C incubator.

### Antibiotic exposure
#### Antibiotics

Ampicillin (Acros Organics Cat. No.: 61177), Streptomycin (Alfa Aesar, Cat. No.: J61299), and Neomycin (Alfa Aesar, Cat. No.: J61499) were all dissolved in deionized water to a working concentration of 20 mg mL$^{-1}$ and sterile filtered. Rifampicin (Tokyo Chemical Industry, Cat. No.: 236-312-0) was dissolved in dimethyl sulfoxide (DMSO) to 20 mg mL$^{-1}$ and subsequently sterile filtered. To measure potential effects of this additional solvent, an equivalent amount of DMSO was added to 15 ppt seawater as a solvent control at the concentration of antibiotic treatment (0.1 and 0.4%, volume/volume). Antibiotic exposures were divided into low (50 $\mu$g mL$^{-1}$) and high (200 $\mu$g mL$^{-1}$) concentrations. The "antibiotic cocktail" is a mix of all four antibiotics at a concentration of 50 $\mu$g mL$^{-1}$ each.

### Larval settlement

Embryos dissociated from the gelatinous egg mass after 24–36 h and were transferred to 24-well plates containing one mL of 15 ppt seawater (control) or one mL of seawater containing DMSO or antibiotics through the observation period. Each treatment had four replicate wells where five embryos were placed into each well. Developing embryos were scored daily for survival and stage of development (embryo, larva, juvenile). Observations ceased once all larvae reached the juvenile stage or stagnated in development for 4 days.

### Culturing bacteria following antibiotic treatment

Juvenile *N. vectensis* ($n = 9$) were exposed to the respective antibiotic treatments per condition for 24 h. Three sets of two juveniles from each treatment condition and controls were washed in sterile 15 ppt ASW to remove residual antibiotics. Anemones were incubated in 10 mL of estuarine broth (EB) to determine if culturable bacteria remained associated with the anemones (NaCl 12.36 g mL$^{-1}$, KCl 0.34 g mL$^{-1}$, CaCl$_2$ 0.68 g mL$^{-1}$, MgCl$_2$ 2.33 g mL$^{-1}$, MgSO$_4$ 3.15 g mL$^{-1}$, NaHCO$_3$ 0.18 g mL$^{-1}$, Peptone 10 g mL$^{-1}$, Yeast Extract 3 g mL$^{-1}$). Broth cultures containing the juveniles were shaken at 200 rpm at 25 °C and checked daily over 7 days for growth of bacteria *via* turbidity. Turbidity of cultures was measured with a spectrophotometer (OD$_{600}$ >1.0, Thermo Fisher Scientific Genesys 30; Thermo Fisher Scientific, Waltham, MA, USA), using the raw, unadjusted measurements. The background absorbance of the Estuarine Broth and borosilicate tubes was OD$_{600}$ = 0.1. Two mL of turbid culture were centrifuged in a microcentrifuge at 8,000× g for 5 min, and the pellet was frozen for DNA extraction and 16S rDNA gene sequencing.

### Generation and maintenance of antibiotic treated Nematostella vectensis

Adult anemones ($n = 4$) were incubated in three conditions to determine how acute and chronic exposure to antibiotics impacts gene expression: (1) No Treatment (NT): 15 ppt ASW with no application of antibiotics, (2) Antibiotic Acute (ABA): antibiotics applied for three days and removed from antibiotics for five days before animal preservation, (3) Antibiotic Constant (ABC): continuous application of antibiotics (8 days) until preservation. The antibiotic solution consisted of 50 μg mL$^{-1}$ of ampicillin, streptomycin, neomycin, and rifampicin. All antibiotics were mixed into sterile filtered 15 ppt ASW.

Anemones in all treatments were fed freshly hatched *Artemia salina* three times per week. In order to eliminate the transfer of bacteria from the *A. salina* to the anemones, cysts were cultured following previously established methods (*Krueger, Shore & Reitzel, 2022*). *Artemia salina* were washed in filter sterilized 30 ppt ASW and confirmed to not harbor culturable bacteria (turbidity test in EB) prior to their use as food.

## Molecular approaches
### 16S rDNA gene amplification

For the broth cultures of the antibiotic exposed juveniles, DNA was isolated from the pelleted bacteria with Thermo Fisher Scientific GENEJet Genomic Purification kit (Cat. #: K0721). The V3-V4 region of the 16S rDNA gene was amplified *via* PCR (Q5 polymerase, New England Biolabs, Cat. #: M0491S) with the primers 5′ TCGTCGGCAGCGTCAGATG-

TGTATAAGAGACAG CCTACGGGNGGCWGCAG 3′ and 5′ GTCTCGTGGGCTCGGA-GATGTGTATAAGAGAC AGGACTACHVGGGTATCTAATCC 3′ (Eurofins). Samples were spot-checked for amplification using gel electrophoresis. Amplified DNA was quantified with a NanoDrop 2000 spectrophotometer (Thermo Fisher Scientific) and normalized to 15 ng $\mu L^{-1}$. Products were purified with the Axygen AxyPrep Mag PCR Clean-up Kit (Cat. #: MAG-PCR-CL-5; Axygen Scientific, Union City, CA, USA), indexed using the Nextera XT Index Kit V2 (Illumina), and then purified again with the same kit. Fluorometric quantification of the final amplicons was performed using Qubit and libraries were checked *via* Bioanalyzer (Agilent Technologies, Santa Clara, CA, USA). Illumina MiSeq sequencing (v3, 2 × 300 bp paired-end reads) was performed at the University of North Carolina at Charlotte.

### RNA extraction

RNA was isolated from each of the preserved anemones (4 biological replicates × 3 experimental groups = 12 extractions) with the RNAqueous kit (Cat. #: AM1912; Ambion, Austin, TX, USA) following the manufacturer's protocol. Following the removal of RNAlater, animals were lysed through pipetting in lysis buffer for <2 min, followed by 2–3 washes and eluted on a column. Genomic DNA was removed using the DNA-free kit (Cat. #: AM1906; Invitrogen, Waltham, MA, USA), and RNA concentration was quantified with a NanoDrop 2000 spectrophotometer (Thermo Fisher Scientific). The RNA tag-based library was prepared at the University of Texas at Austin's Genomic Sequencing and Analysis Facility (GSAF) as in *Meyer, Aglyamova & Matz (2011)* and sequenced with Illumina HiSeq 2500 (100 base single end). Briefly, total RNA was transcribed to cDNA and subsequently purified with AMPure XP beads (Cat. #: A63880; Beckman Coulter, Brea, CA, USA). Samples were amplified with 18 PCR cycles and unique Illumina barcodes were added to the amplicons for indexing. After an additional purification step, libraries were pooled, spot-checked for quality on a Bioanalyzer (Agilent and Pico) and size-selected using BluePippin (350–550 bp fragments) prior to sequencing.

## Bioinformatic analyses
### 16S rDNA gene classification

Raw reads and quality scores were imported into QIIME2 for analysis (*Bolyen et al., 2019*). The demultiplexed sequences were trimmed to 250 bases using the median quality scores of 19 generated in QIIME2, while maintaining sequence length. Forward read quality at base 250 had 25th and 75th percentile quality of 34 and 38, respectively. Reverse read quality trimmed at 250 bases had 25th and 75th percentile quality of 15 and 31 respectively. The number of reads retained ranged from a minimum of 10,884 counts, maximum of 27,080, and median frequency of 19,959 reads. The taxonomies of the amplicon sequence variants (ASVs) were assigned with a Naive Bayes classifier with the SILVA database (*Quast et al., 2012*) trained for 16S V3/V4 primers used to amplify the region. ASVs with less than 10 total counts were removed from the dataset. Lastly, a PCA plot and heatmap of the respective communities were imported and constructed in the ampvis2 R package (*Andersen et al., 2018*), and the taxonomic distributions were plotted as bar charts using default parameters.

*Differential gene expression*

Raw sequences were adapter clipped and trimmed using cutadapt (*Meyer, Aglyamova & Matz, 2011*). Next, the trimmed sequences were aligned to the *N. vectensis* Vienna transcriptome V2 (see Data Availability statement) using Bowtie2 (*Langmead & Salzberg, 2012*). The alignment sam file was converted to the bam file type with the package samtools (*Li et al., 2009*). The bam files were then used to generate a counts table with the featureCounts package (*Liao, Smyth & Shi, 2019*). Differentially expressed genes (DEGs) were identified using $padj \leq 0.05$ (Wald statistic; DESeq2) and log-fold change greater than 1.5. DEGs were compared and graphed across treatments with the R package UpSetR (*Conway, Lex & Gehlenborg, 2017*; *Love, Huber & Anders, 2014*). The rlog R function transformed counts and performed a PCA based on Manhattan distances. Statistical significance of the Manhattan distances was calculated using adonis2 in the vegan R package (*Dixon, 2003*). PCA of the total gene set was analyzed with prcomp and viewed with ggplot2 in R (*Wickham, 2016*). Gene Ontology (GO) terms were compared for over- or underrepresentation in the DEGs using GO_MWU with log-fold change values (*Wright et al., 2015*). Shared and unique GO terms across the treatments were compared with the R package UpSetR (*Conway, Lex & Gehlenborg, 2017*) and generated with ComplexHeatmap (*Gu, Eils & Schlesner, 2016*).

# RESULTS

## Antibiotics extend time to larval settlement

To determine the impact of antibiotics on the development and settlement of *N. vectensis*, embryos were exposed at two concentrations and the length of time in days was measured to reach the juvenile stage. Embryos and larvae exhibited no observed morphological changes due to exposure to any antibiotic individually or in combination. Developing larvae took longer to settle and enter the juvenile stage when exposed to most of the antibiotic treatments. The Development Time (DT) for 50% of larvae in the combined antibiotic treatment to reach the juvenile stage (DT50, measured in days) increased by 3 days (+50% longer compared to control larvae) (Fig. 1A). Additionally, the DT100 (days required for 100% of the larvae to settle) showed a similar delayed settlement time for larvae cultured with most antibiotics (Fig. 1B). The DT50 of larvae in DMSO at 1% v/v was the only condition that was not statistically different from control (one-way ANOVA, $p = 0.666$). When comparing the settlement time for 100% of larvae, ampicillin at 200 µg mL$^{-1}$, was the only condition that was not statistically different compared with controls (one-way ANOVA, $p = 0.125$), while the remaining treatments had significantly longer time to settlement (Table 1).

## Bacteria are cultivatable following removal of antibiotics

Juveniles removed from an antibiotic treatment and cultured in media generally resulted in cultivated bacteria in most conditions over the following week. Turbidity measurements were taken daily for seven days where the total number of recovered cultures (Fig. 2A). Bacteria grew most quickly in controls ($n = 8$), DMSO (50 µg mL$^{-1}$, $n = 9$, 200 µg mL$^{-1}$, $n = 6$), and ampicillin treated cultures (50 µg mL$^{-1}$, $n = 9$, 200 µg mL$^{-1}$, $n = 8$), which all

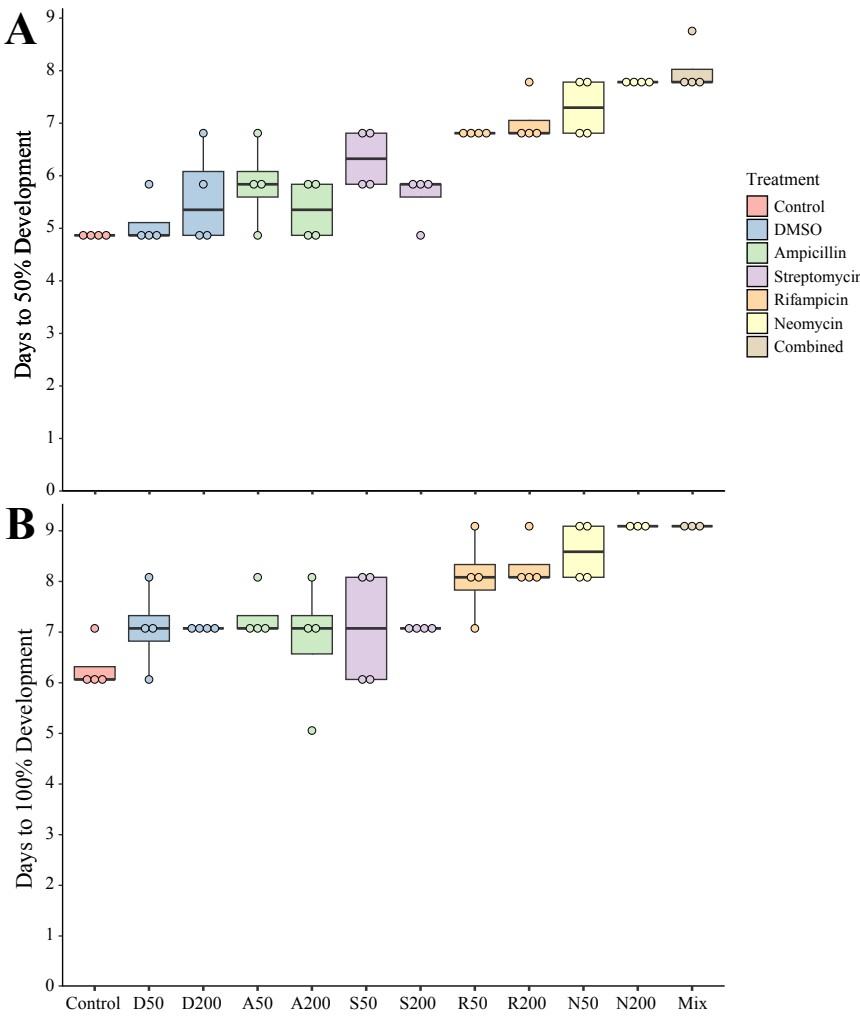

**Figure 1** **Settlement time of newly spawned *N. vectensis*.** (A) The number of days required for 50% of the population ($n = 20$) to settle. (B) The number of days required for 100% of the population ($n = 20$) to settle. Means are represented by the thick line, boxes represent the 25% and 75% quartiles, and the vertical lines indicate the remaining quartiles. Statistical values can be found in Table 1.

became turbid ($OD_{600}$ greater than 1.0) within 3 days. Treatments with other antibiotics resulted in delayed growth of bacteria. Streptomycin at 50 ($n = 6$) and 200 $\mu g\ mL^{-1}$ ($n = 8$), were significantly delayed when compared to controls ($p = 0.0078$, $p = 0.0229$, respectively). Rifampicin 50 $\mu g\ mL^{-1}$ ($n = 5$) and at 200 $\mu g\ mL^{-1}$ ($n = 2$) and neomycin at both 50 $\mu g\ mL^{-1}$ ($n = 8$) and 200 $\mu g\ mL^{-1}$ ($n = 5$) resulted in significantly longer times to turbidity compared to the control ($p = 0.0427$, $p = 0.0004$, $p = 0.0002$, $p < 0.0001$, respectively). The antibiotic cocktail (Mix) yielded growth in three of the nine replicates, which was significantly longer when compared to the controls ($p < 0.0001$).

Sequencing of the bacteria that grew in each antibiotic treatment resulted in a total of 359 ASVs across all conditions. The total recovered ASVs is likely to be a reduced representation of the total culturable microbial community, and may favor fastidious organisms. After

**Table 1 Statistical results from DT50 and DT100 settlement assays.** Bold font denotes statistical significance with *p*-values less than 0.05.

| Settlement *p*-values | DT50 | DT100 |
|---|---|---|
| Control *vs.* DMSO50 | 0.6655 | **0.0039** |
| Control *vs.* DMSO200 | **0.0002** | **0.0039** |
| Control *vs.* Amp50 | **<0.0001** | **<0.0001** |
| Control *vs.* Amp200 | **0.0352** | 0.1251 |
| Control *vs.* Strep50 | **<0.0001** | **0.0039** |
| Control *vs.* Strep200 | **0.0002** | **0.0039** |
| Control *vs.* Rif50 | **<0.0001** | **0.0069** |
| Control *vs.* Rif200 | **<0.0001** | **<0.0001** |
| Control *vs.* Neo50 | **<0.0001** | **<0.0001** |
| Control *vs.* Neo200 | **<0.0001** | **<0.0001** |
| Control *vs.* Combined | **<0.0001** | **<0.0001** |

filtering features that had less than 10 counts in the total dataset, 150 ASVs remained. The alpha diversity (Faith's Phylogenetic Diversity; Kruskal-Wallace) across antibiotics revealed statistical differences between eight pairwise comparisons, including A200 relative to S50 ($p = 0.003$), A50 ($p = 0.007$), N200 ($p = 0.019$), and N50 ($p = 0.019$) (Fig. 3A). Treatment constituted 15.3% of the variation, but was not significant (Bray–Curtis, $p = 0.675$). The dominant taxonomic class across all samples was Gammaproteobacteria, constituting 37.9% of bacteria over all conditions. The lowest ASV diversity across treatments was DMSO and the highest diversity was in the streptomycin treatment at 200 $\mu$g mL$^{-1}$ (Faith PD 0.72, 3.05, respectively; Fig. 2B). The most, and third most abundant ASVs across all samples were both from the genus *Microbacterium* and were present in 38 and 35 samples respectively. The second most prominent ASV was a *Vibrio* species and was present in 24 of the 69 anemones that exhibited growth. A NMDS plot showed the communities of these ASVs that persisted across treatments remain similar (Fig. S1). The highest abundance order of bacteria belongs to Micrococcales, Flavobacteriales, and Vibrionales, respectively (Fig. 3A). Additionally, streptomycin treatments resulted in the most ASVs of all treatments, while the condition with all antibiotics resulted in the least number of ASVs (Fig. S2). Eleven ASVs were consistent across all conditions including Micrococcales, Vibrionales, Flavobacteriales, Sphingomonadales, Oceanospirallales, Psuedomonadales, and Cytophagales. Moreover, we found several anemones to be represented by a singular ASV across many antibiotic exposures (Fig. 3B).

## Differential gene expression with antibiotic exposure

The impact of acute and constant doses of antibiotics on *N. vectensis* gene expression revealed significant differences in differential gene expression between each treatment. The principal component analysis revealed clustering of replicates from each of the three groups (No Treatment (NT), Antibiotic Acute (ABA), and Antibiotic Constant (ABC)), where the first two components represented 25.7% of the variation (Fig. 4A). Additionally, the Manhattan distances were statistically different (PERMANOVA, $p = 0.001$), and Condition explained 25.9% of the variation ($R^2 = 0.259$).

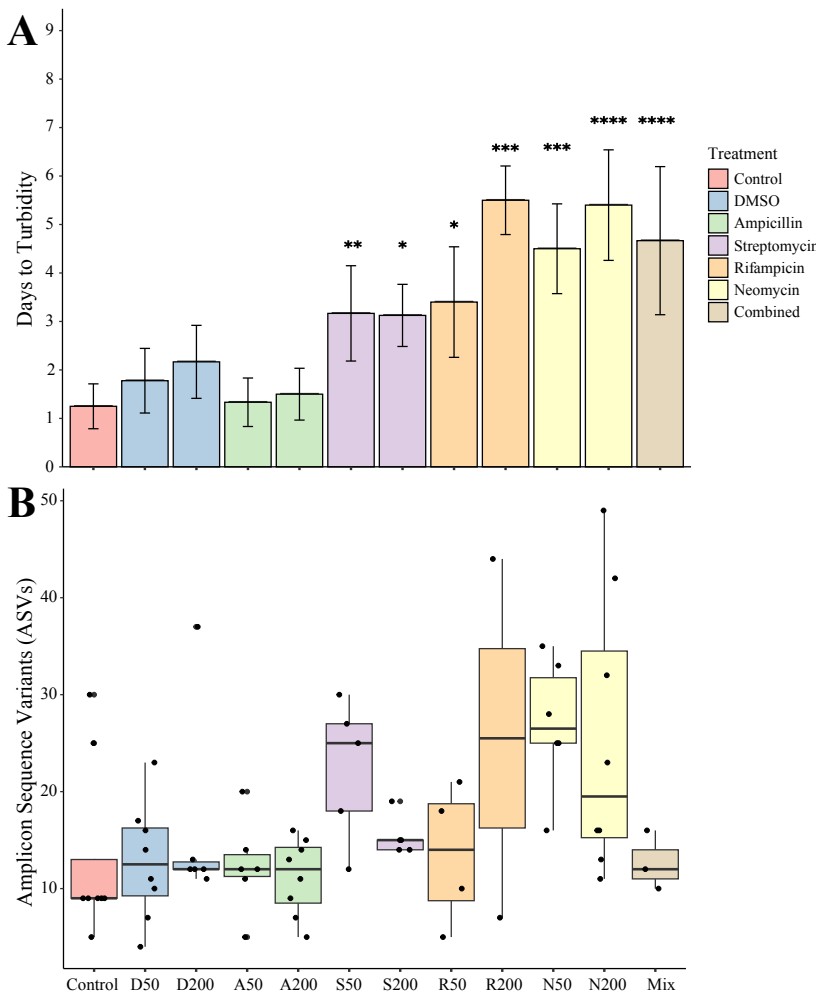

**Figure 2** **Growth and amplicon sequence variant distribution in *Nematostella vectensis*.** (A) Days to turbidity. Bars represent the mean and error bars represent the standard deviation. Asterisks represent statistical significance compared to controls (* $p < 0.05$; ** $p < 0.01$, *** $p < 0.001$; **** $p < 0.0001$) (B) Distribution of amplicon sequence variants (ASVs) across treatments.

*Nematostella vectensis* that were acutely treated with antibiotics resulted in 46 differentially expressed genes (19 up-regulated, 27 down-regulated) when compared to controls (Fig. 4B; Fig. S3). The highest up-regulated gene in the acute antibiotic treated animals was CpG-binding protein, a gene involved in genome methylation (*Wang et al., 2014*). Cyclin B was upregulated in the acute treatment condition (mitosis, reviewed in: *Dorée & Galas (1994)*. Additionally, SoxC was upregulated compared to the controls (transcription factor; neurogenesis; (*Magie, Pang & Martindale, 2005*; *Richards & Rentzsch, 2015*; *Steger et al., 2022*). The 28S ribosomal protein S16 was upregulated in the acutely treated anemones. The most down-regulated differentially expressed gene was a histone ubiquitination protein DZIP3 (*Inoue et al., 2015*). Other genes with lower expression in the acute antibiotic condition included two low-density lipoprotein (LDL) receptors, tubulin

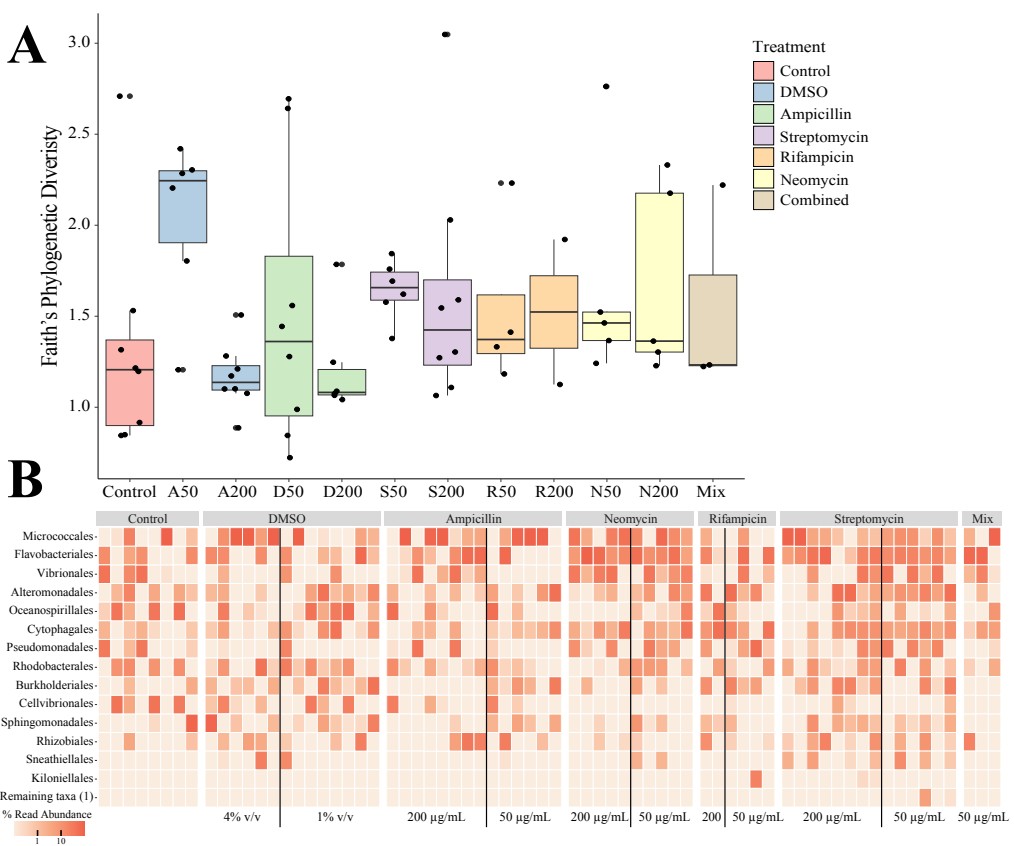

**Figure 3** **Persistent bacteria in juvenile *N. vectensis* following antibiotic exposure.** (A) Faith's phylogenetic diversity. Bars represent the mean and error bars represent the standard deviation. (B) Heatmap of individual samples categorized into family level taxonomy.

beta-4B and three genes related to neuronal function (synaptotagmin-4, nectin precursor, neuropeptide FF receptor-1).

Constant application of antibiotics resulted in 231 DEGs (116 up-regulated, 115 down-regulated) compared to the controls, which include chaperones, mitochondrial genes, and transcription factors (Fig. S3). Twenty-one of the 231 genes were shared with the DEGs expressed in the acute *versus* control comparison (Fig. 4B), which include DZIP3, ubiquitin, and LDL receptors. The most upregulated gene in the constant application of antibiotics treatment was peroxisomal membrane protein 11B. Additionally, a predicted failed axon connectors homolog was significantly upregulated in the animals exposed to antibiotics. Several genes related to mitochondrial function were also significantly upregulated including Cytochrome C, SCO cytochrome oxidase deficient homolog 1, rRNA methyltransferase 1, mitochondrial precursor, and Enoyl-CoA delta isomerase 1. Five genes related to chaperone function (calumenin-A precursor, two calumenin-B precursors and, a predicted DNAJ homolog, and PDIA4) were upregulated in the constant antibiotic treatment. Two transcription factors, OTX and HIF-prolyl hydroxylase PH-4-like protein, were upregulated in the constant antibiotic exposure. Three genes (tubulin beta-4B

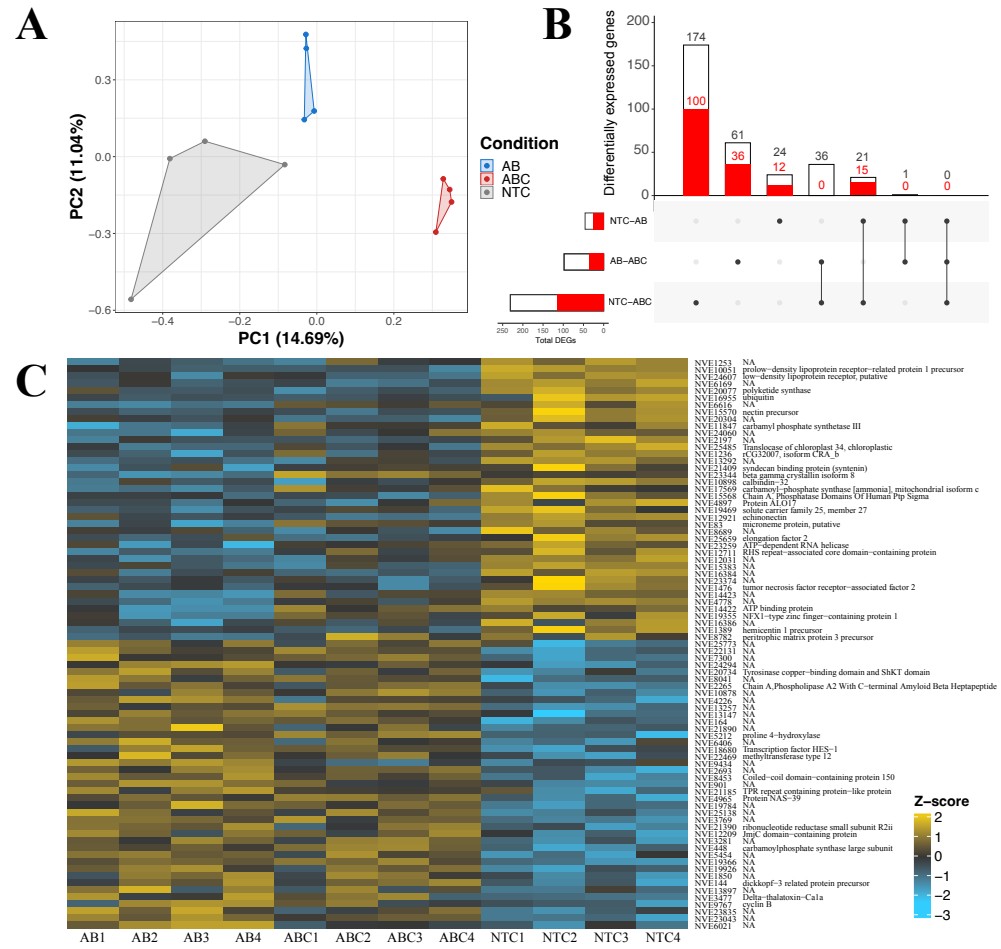

**Figure 4** **Differential gene expression in *Nematostella vectensis*.** (A) PCA of all genes in across control, antibiotic acute, and antibiotic constant conditions. The first two components constitute 25.7% of variance in the data (PERMANOVA; $p = 0.001$). (B) Upset diagram of the differentially expressed genes across conditions. White bars represent genes present in the entire set, while red bars indicate the total number of down-regulated genes in each set. (C) Heatmap of the top 40 greatest differentially expressed genes. P-adjusted values were less than 0.05 and the base means were greater than 150. NA indicates no annotation for the respective gene. (NTC, No Treatment Control; AB, Antibiotic Acute; ABC, Antibiotic Constant).

chain protein, two LDL receptors and DZIP3) were downregulated in constant antibiotic conditions as also observed in acute exposures. An additional LDL receptor related protein was downregulated under the constant antibiotic condition. The metabolic genes malate dehydrogenase 1 and glucose-6-phosphate dehydrogenase isoform B was downregulated in the antibiotic treated condition. Furthermore, two mitochondria related genes were downregulated, isochorismatase domain-containing protein 2 and cytochrome P450, family 1, subfamily C, polypeptide 2.

Comparisons of gene expression between acute antibiotic and the constant antibiotic treatment resulted in 98 DEGs (62 upregulated, 36 downregulated; Fig. S3). These genes included mitochondrial genes and zinc-finger proteins. Thirty-six of these differentially

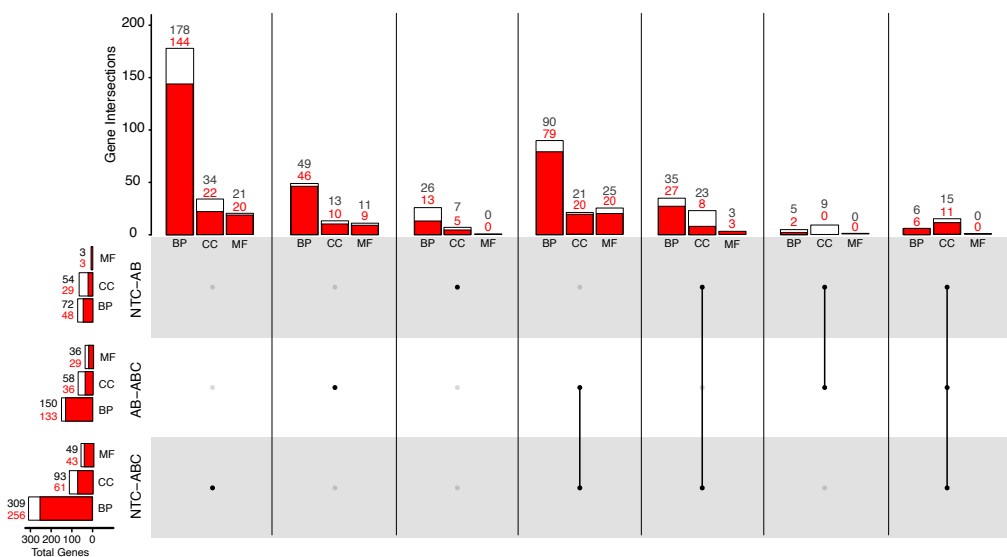

**Figure 5** **Distribution of Gene Ontology terms across treatments.** White bars represent total significant terms (padj < 0.05), while red bars represent the depleted terms in each set. Groups are split into Biological Process (BP), Cellular Component (CC), and Molecular Function (MF). (NTC, No Treatment Control, AB, Antibiotic Acute; ABC, Antibiotic Constant).

expressed genes were shared with the control *versus* constant application comparison including Cytochrome C, while none were shared with the acute *versus* control comparison (Fig. 4B). The most upregulated DEG in the constant antibiotic treatment condition was thiosulfate sulfurtransferase-like protein, in addition to four other genes related to mitochondrial function were upregulated: Cytochrome C, 2-oxoisovalerate dehydrogenase subunit alpha, rhodanese-like protein, and Cytochrome P450 3A2. Additionally, arginyl-tRNA synthetase was upregulated in the constant application condition. Among the downregulated genes were the ES1 mitochondrial protein, tubulin alpha chain, solute carrier family 7, member 4 and solute carrier family 1, member 3a.

## Gene ontology comparison of DEGs

A total of 72 biological process (BP; 24 enriched, 48 depleted) Gene Ontology (GO) terms were identified in DEGs between the acutely treated animals compared to the control animals (Fig. 5). Mann–Whitney U (MWU) rank sum test showed a depletion of GO terms related to "cellular response to oxygen-containing compound" (Fig. S4). Three terms categorized as molecular function (MF; zero enriched, three depleted) related to the depletion of "signaling receptor activity" were identified. Fifty-four cellular component (CC; 30 enriched, 24 depleted)) GO terms were significant in the antibiotic acute condition compared to the control. Three terms were found to represent CC, including the enrichment of "endoplasmic reticulum lumen", "spindle", and the depletion of "neuron projection" terms.

Comparatively, 309 BP GO terms were significant (256 enriched, 53 depleted) in animals that were maintained under constant antibiotic stress relative to controls (Fig. 5).

Of the terms categorized as BP, terms related to "DNA integrity checkpoint", "RNA processing", "protein folding", and "genitalia development" were enriched, while terms related to "multicellular organismal signaling", "cyclic nucleotide metabolic process", and "regulation of locomotion" were depleted (Fig. S4). Forty-nine MF terms (six enriched, 43 depleted) were significant which are related to the enrichment of "endopeptidase activity" and "unfolded protein binding", while depleted terms were related to "amino acid transmembrane transporter activity" and "cell adhesion molecule binding". Ninety-three CC terms (32 enriched, 61 depleted) were related to the enrichment of "endopeptidase complex", "nucleolus", "organelle inner membrane", and "chromosomal region", while depleted terms were related to "cilium" and "cell junction".

When comparing the antibiotic acute and antibiotic constant treatments, a total of 150 significant BP GO terms were identified (17 enriched, 133 depleted; Fig. 5). Of these terms, enrichment of "antigen processing and presentation of peptide antigen *via* MHC class I", "RNA processing", and "cuticle chitin metabolic process" terms occurred, while terms related to "drug transport", "microtubule-based movement", and "circulatory system development" were depleted (Fig. S4). Thirty-six MF GO terms (seven enriched, 29 depleted) were significant when comparing the antibiotic acute and antibiotic constant conditions, where enriched terms function as "endopeptidase activity" and "catalytic activity, acting on tRNA", and depleted terms function as "cytoskeletal protein binding" and "amino acid transmembrane transporter activity". Lastly, 58 significant CC GO terms (17 enriched, 41 depleted) include the enrichment of "vacuole", "endopeptidase complex", "P-body", "mitochondrial envelope", while depleted terms related to "cilium" and "extracellular matrix".

## DISCUSSION

Antibiotics are commonly used to reduce or eliminate bacteria associated with animals and other hosts. These compounds will differentially impact portions of the bacterial community depending on the antibiotic and the relative susceptibility of the bacteria to each individual antibiotic and the respective concentration. In addition, antibiotics can have diverse unintended impacts on animals and other eukaryotes due to off-target effects. Here, we found that exposure to antibiotics of different classes and concentrations impacts both *Nematostella vectensis* and the associated bacteria.

Although treatment with antibiotics delayed the growth of bacteria associated with *N. vectensis*, we found a diverse community of bacteria were culturable after antibiotic exposure. When anemones are exposed to a combination of antibiotics, some of the tested animals resulted in growth in liquid growth media. For these, the growth was delayed compared to controls. This may be an indication that some bacteria may have persisted in the animals tested for transcriptional responses (see below), though the high nutrient media was an optimal environment for bacterial growth. The media may preferentially select for faster growing species present in this community compared to other approaches (*e.g.*, agar plates). Despite this potential bias, we still identified 150 ASVs using this method. While this number is fewer than comparable studies with *N. vectensis* using

only sequence-based approaches from anemone homogenates, *e.g.*, (*Baldassarre, Reitzel & Fraune, 2023*; *Mortzfeld et al., 2016*), the identified ASVs were taxonomically diverse. The largest fraction (37.9%) across all conditions belong to the Gammaproteobacteria clade. Additionally, other clades grew in media following removal of antibiotics included ASVs from Rhodobacterales and Alteromonadales. This culturable fraction of the bacteria from *N. vectensis* adults exposed to antibiotic was similar to previous results with the coral *Pocillopora damicornis* (*Connelly et al., 2022*). Adult corals exposed to antibiotics showed significant changes in the bacterial community, with species from the families Rhodobacteraceae and Alteromonadaceae still present following treatment (*Connelly et al., 2022*). Similarly, in another coral species, *Euphyillia paradivisa*, antibiotic treatment resulted in Alphaproteobacteria instead of Gammaproteobacteria as the most common order of bacteria, followed by Bacteriodes (*Meron et al., 2020*). Gammaproteobacteria were still persistent in *Euphyllia* and included Alteromondales, Oceanspiralles, and Vibrionales, all of which were detected in our study with *N. vectensis*.

The potential for bacteria to survive treatments *via* mechanisms such as antibiotic resistance (*Uddin et al., 2021*), biofilm formation (*Antunes, Leao & Vasconcelos, 2018*; *Armstrong et al., 2001*; *Schillaci et al., 2010*) or dormancy (*Mu et al., 2020*; *Wang et al., 2021*; *Zhang et al., 2021*) can provide methods of survival for a fraction of the microbial community. These mechanisms may hinder some efforts to eliminate pathogens or create axenic individuals for microbiome manipulations. Here, we found that some bacteria were not cultivatable following particular antibiotic treatments. For instance, Oceanospiralles was prevalent across most conditions, but was not detected in the anemones that were exposed to 200 ug mL$^{-1}$ of neomycin, but was detectable at 50 ug mL$^{-1}$. Additionally, ASVs from the Burkholderiales order were widely cultivatable except in the anemones that were exposed to 200 ug mL$^{-1}$ of ampicillin and the mixture of all antibiotics. The relative sensitivity of a portion of the microbiome to higher concentrations of antibiotics suggest even resistant bacteria can be eliminated *via* increases in concentration or introduction of multiple antibiotics that target different prokaryotic mechanisms. Moreover, ASVs that belong to the Cellvibrionales order were cultivated across most conditions except for neomycin at both concentrations and the mixture of antibiotics. Lastly, we found that rifampicin and the mixture of antibiotics were effective at eliminating the microbiome, as evidenced by growth in the fewest number of exposed anemones. Rifampicin inhibits RNA synthesis in bacteria (*McClure & Cech, 1978*). This mechanism of action may be particularly effective at eliminating bacteria in host organisms in this or other coastal species environments and requires further characterization with other animal hosts.

Unexpectedly we observed a higher diversity of ASVs in some antibiotic treatments when compared with the unexposed control anemones. Indeed, the highest diversity of 49 ASVs were detected in the high concentration of streptomycin. As stated earlier, we hypothesize that the elimination of the some fast growing organisms from the high dosage of streptomycin or other antibiotics may allow for growth of rarer bacteria, which could result in greater quantity of ASVs when compared to controls. These persisting bacteria may constitute the initial bacteria that then compose the microbiome of *N. vectensis* following these relatively short duration treatments. Our method provides insight into
readily culturable bacteria following antibiotic treatment, which would include those readily able to grow and potentially populate an organism with a depleted microbiome.

Antibiotic exposures resulted in distinct transcriptional responses in adult *N. vectensis* exposed to short term and constant treatments. The transcriptional response suggests that a diverse array of cellular processes and functions are impacted. The constant antibiotic exposure resulted in more differentially expressed genes (DEGs) when compared with no treatment conditions and the acute exposure. The predicted functions for the DEGs included mitochondrial functions, metabolism, and cellular components such as tubulin and actin. Additionally, endoribonuclease and immunity related genes in were differentially expressed similar to a previous study with the coral *Pocillopora* (*Connelly et al., 2022*). The depletion of various processes such as neural, ion transport, and organelle cellular components indicates the animal experiences a variety of sublethal effects that have been described in other systems (*Connelly et al., 2022*; *Meron et al., 2020*). Moreover, it was recently shown that chitin metabolism pathways are regulated by the microbiome, which is consistent with the gene ontology analysis of the antibiotic constant condition (*Domin et al., 2023*). One of the upregulated transcripts in the antibiotic constant treatment was Cytochrome C, which is central for mitochondrial function for ATP synthesis and apoptosis, suggesting that antibiotics can target metabolic processes within the host cells (*Miller & Singer, 2022*). While antibiotic off target effects on Cytochrome C has not been described in *N. vectensis,* exposure of *Oryzias latipes* and *Danio rerio* to erythromycin, a macrolide, showed downregulation of Cytochrome C, as well as a variety of other metabolism-related genes, reduced locomotion, and energy metabolism (*Li & Zhang, 2020*). Here, we describe an extension of development due to antibiotic exposure, which may be related to metabolic effects, and is supported by gene expression data found in adult *N. vectensis*. Oxidative stress has been identified in *Sparus aurata* following exposure to antibiotics, and has significant effect on the physiology of the organism including DNA damage. (*Rodrigues et al., 2019*). The effect of constant antibiotic application has a variety of transcriptional effects on the host organism, which may exert physiological effects on the anemones during these prescribed treatments.

When *N. vectensis* was exposed to antibiotics for a short duration, and subsequently cultured in sterile seawater, transcriptional effects were still detectable after five days. This short duration of antibiotic exposure resulted in a decrease in expression of genes involved with the nervous system. SoxC is a transcriptional factor that has a primary role in differentiation of progenitor cells into neurons, gland cells, and cnidocytes, which was downregulated in the acute treatment condition (*Kavyanifar, Turan & Lie, 2018*; *Steger et al., 2022*). Additionally, OTX has also been described to promote nervous system development (*Cañestro, Bassham & Postlethwait, 2008*; *Mazza et al., 2007*). While our experiment does not allow us to separate the direct effects of a missing portion of the microbiome due to the antibiotic exposure and some persisting bacteria, the differential expression of these core neural genes suggests that the nervous system of *N. vectensis* may be improperly regulated. The microbiome is an important contributor to the development and function of the nervous system in other animals (*Jameson et al., 2020*;

*Sharon et al., 2016*). Thus, this result suggests that more attention on the intersection of microbes and the sea anemone nervous system may be informative in future studies.

DZIP3, the most downregulated gene in the acute antibiotic treatment, has been identified to regulate the immune response to lipopolysaccharide in the oyster *Crassostrea gigas* (*Cheng et al., 2016*). Additionally, a zinc finger protein, which may be involved in limiting viral RNA replication (*Esposito et al., 2022*; *Wang et al., 2019*), was downregulated in this constant application condition. These results could indicate two possibilities. Antibiotics could directly impact the immune system, resulting in downregulation of these genes. Alternatively, the antibiotics may reduce or remove the bacteria from the animal and indirectly reduce expression of immune genes by reducing the burden of bacteria within the host. Resolving the direct and indirect effects of antibiotic exposure is important for deciphering the mechanisms of animal responses to different classes of antibiotics.

Antibiotics of multiple classes extended the developmental time for this species. In larvae, antibiotic exposure increased the time to settlement by as much as 50%. The impact of certain microbes and their settlement cues have been characterized in several systems (*Tran, 2022*). For example, *Pseudoalteromonas* spp. produces tetrabtomopyrrole (TBP) which can induce settlement in a wide range of marine invertebrates including *Heliocidaris erythrogramma* (*Huggett et al., 2006*), *Porites astreoides, Orbicella franksi,* and *Acropora* spp. (*Sneed et al., 2014*; *Tebben et al., 2011*). The combination of a depletion of beneficial bacteria and indirect effects of antibiotic exposure may delay settlement of larvae. While longer settlement times does not induce mortality alone, the increase in time may restrict the larvae from successful settlement and may result in larval mortality from predation due to longer times in the water column. In *N. vectensis*, the bacterial community shifts through metamorphosis (*Mortzfeld et al., 2016*), indicating the interactions with the host may change concurrently, although bacteria specific for settlement cues are unknown.

Taken together, we found in this study that there are broad, non-lethal effects of antibiotic exposures on the cnidarian *Nematostella vectensis*. Gene expression data suggest that many functions, including metabolism, mitochondrial function, and neural processes may be impacted. Moreover, we found that antibiotics slow the development process in *N. vectensis*, which may be due to the potential for reduced metabolism and cell differentiation (*e.g.,* SoxC for differentiation of progenitor cells). Additionally, the recovery of cultivatable bacteria in all conditions indicates that antibiotic exposures may require longer durations and/or different combinations to consistently eliminate all bacteria.

## CONCLUSIONS

We investigated the effects of antibiotics on the model cnidarian *Nematostella vectensis* and the associated bacterial community. We found in adult anemones decreased expression of genes and many GO terms (*i.e.,* mitochondrial, metabolism, and development) were depleted with both the antibiotic acute and constant conditions. If these processes are reduced in capacity, it may reduce the fitness of the organism. Additionally, some bacteria survive and proliferate after antibiotic treatment, thus indicating antibiotic treatments as shown in this work may not successfully create axenic organisms in some cases.

Interestingly, the settlement time of the larvae increased under many of the differing antibiotic conditions. Moreover, the transcriptional effects on the host vary based on duration of antibiotic exposure and some impacts extend for days after removal of the antibiotics. Together, these results reveal a multi-faceted impact of antibiotics on a marine invertebrate and its bacterial community that are important to consider in future applications in laboratory and field experiments.

### Funding

Quinton Krueger and Adam Reitzel were supported through Human Frontiers Research Program award RGY0079/2106. Research reported in this study was also supported by NIH award R15GM137253 and NSF Award 2044826 to Adam Reitzel. The funders had no role in study design, data collection and analysis, decision to publish, or preparation of the manuscript.

### Grant Disclosures

The following grant information was disclosed by the authors:
Human Frontiers Research Program Award: RGY0079/2106.
NIH Award: R15GM137253.
NSF Award: 2044826.

### Competing Interests

The authors declare there are no competing interests.

### Author Contributions

- Quinton Krueger conceived and designed the experiments, performed the experiments, analyzed the data, prepared figures and/or tables, authored or reviewed drafts of the article, and approved the final draft.
- Britney Phippen conceived and designed the experiments, performed the experiments, authored or reviewed drafts of the article, and approved the final draft.
- Adam Reitzel conceived and designed the experiments, authored or reviewed drafts of the article, and approved the final draft.

### Data Availability

The raw sequences used for taxonomic abundance analysis of 16S rDNA gene markers are available at Zenodo: Krueger, Q., & Reitzel, A. (2023). Antibiotics alter development and gene expression in the model cnidarian *Nematostella vectensis*. https://doi.org/10.5281/zenodo.8403880.

The raw sequences, gene count table, and metadata are available at GEO: GSE167114.

The Nematostella vectensis gene model is available at FigShare: Fredman, David; Michaela Schwaiger; Fabian Rentzsch; Technau, Ulrich (2013). *Nematostella vectensis* transcriptome and gene models v2.0. figshare. Dataset. https://doi.org/10.6084/m9.figshare.807696.v3.

## Supplemental Information

Supplemental information for this article can be found online at http://dx.doi.org/10.7717/peerj.17349#supplemental-information.

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
