# Peer review of "Antibiotics alter development and gene expression in the model cnidarian Nematostella vectensis"

_PeerJ, doi:10.7717/peerj.17349_

## Round 0.1 · original submission · Major Revisions

Please provide a detailed point-by-point rebuttal to all of the reviewers' comments along with your revised manuscript.

Reviewer 1 ·

Basic reporting

The manuscript of Krueger et al. performs antibiotic treatment in the marine model organism Nematostella vectensis and monitored changes in the host associated bacterial community, host development and host gene expression.
My main criticism relates to two points: (i) to the bacterial analysis. For the analyses of the bacterial community a cultivation approach in liquid media was used to monitor and identify remaining active bacteria on the host after antibiotic treatment. This approach introduces a large bias toward fast-growing bacteria and leads to an overestimation of these bacteria in subsequent analyses. The authors should clearly indicate why they have chosen this approach and discuss the advantages, but also the disadvantages compared to direct 16S rRNA gene sequencing on DNA or RNA level or cultivation of bacteria on agar plates. These methods would avoid the overestimation of fast-growing bacteria and the inhibition of slow growing bacteria. (ii) to the design of the experiment. With the performed experimental design direct and indirect effects of antibiotics on host gene expression can not be separated. This should be clearly stated and discussed. Details see below.

Experimental design

Line 306: With this experimental setup, direct or indirect effects of antibiotics on host gene expression cannot be separated. Indirect effects might be mediated by the changed bacterial communities. To separate these effects, germfree animals need to be analyzed with and without antibiotics. This point needs to be discussed. For bacterial effects on Nematostella gene expression see https://www.biorxiv.org/content/10.1101/2022.12.13.520252v1.

Validity of the findings

Line 171: What is the reason for using the threshold of OD>1 for detecting bacterial growth. Many bacteria do not reach an OD of 1.0 in liquid media. Did the authors include a negative control (media with no turbidity) in the subsequent 16S rRNA sequencing?

Line 283: It is not clear to me what conclusions can be drawn from the data of the cultured bacteria. The growth of bacteria in liquid culture leads to an over presence of fast-growing bacteria. Therefore, these data allow only a small conclusion about the bacterial community of the host.

Additional comments

Specific points:

Line 42- 43: here, both general statements require further references based on additional host species.

Line 57: Here the potential effects of antibiotics on eukaryotic organelles (endosymbiont theory) needs to be considered.

Line 91: Costa et al studied the bacterial community of Aiptasia not Nematostella

Line 104: Carrier & Reitel 2019 did not study the microbiome of Nematostella. Instead, you can use also the Baldassarre et al 2021 paper to support this statement.

Line 112: Menard 2018 is not a peer reviewed publication. The authors should clarify in the references that this is a dissertation and should provide a link.

Line 157: not clear, how long embryos were maintained in antibiotics.

Line 169: The authors should clarify the reasons for using liquid cultures vs. agar plates? During growth in liquid media fast growing bacteria overgrow slow growing bacteria and get overestimated.

Line 191: The authors should state, why they sequenced the cultivated bacteria instead of sequencing directly the bacteria associated with Nematostella? If active bacteria are the target of analysis, 16S rRNA-based gene analyses based on RNA could be used.

Line 386: Recently, it was shown that chitin synthase in Nematostella is regulated by bacterial colonizers (see. https://www.biorxiv.org/content/10.1101/2022.12.13.520252v1). This is a strong indication that the observed changes in host gene expression is also influenced by indirect effects, mediated via the microbiome.

Line 440: The observed effects of antibiotics on mitochondria needs to be discussed in the light of the endosymbiont theory (see e.g. https://academic.oup.com/jac/article/77/5/1218/6535933)

Line 454: The regulation of DZIP3 are another indication for indirect effects of antibiotics on host gene expression.

Reviewer 2 ·

Basic reporting

Review:

The study focuses on investigating the effects of antibiotics on the development and settlement of a sea anemone using 16S metabarcoding to assess microbial disruption and gene expression analysis to elucidate the associated processes. The study provides insight into the potential ecological consequences of antibiotic treatments in marine ecosystems, particularly with regard to coral reefs. I recommend publication following major revisions.

Introduction:

In the introductory section, I recommend expanding on the importance of host-microbe interactions within the context of cnidarian biology and life history. Providing examples and referencing existing studies that demonstrate the influence of the microbiome on developmental processes would be more suitable for the introduction.

It may be better to replace the second paragraph with the third to shift the focus towards the side effects of antibiotic applications in marine environments. Beginning with a statement such as "Antibiotic use in aquaculture is projected to grow by more than 30% by 2030 to increase food production worldwide (Schar et al., 2020). However, the non-specific effects of antibiotics on aquatic host organisms remain poorly characterized (Yang et al., 2020)" would provide a more logical progression.

In lines 50-55, could be removed and replaced the content with information on the use of antibiotics for coral disease treatment or antibiotic applications in marine organisms. This could be a better link to potential side effects of antibiotic methods on the life history of other organisms.

Consider removing lines 89-114, as they appear disconnected from the preceding paragraph. While the paragraph discusses the suitability of Nematostella as a model system, it could be integrated somewhere around the last paragraph rather than being presented as a standalone paragraph.

Experimental design

Methods:

Specify whether seawater or deionized water (DI) was used from the outset in lines 150-152. Additionally, include information about the salinity of the treatment for clarity.

Clarify in lines 206-208 whether the animals were pooled for extraction or not.

Validity of the findings

Figures:

Figure 1 (A&B): To improve visualization and interpretation, consider adding error bars to all data points. Additionally, presenting the data as boxplots with overlaid points (geom points) could be more informative.

In Figure 2, include information about the measure of turbidity OD used and provide more information in the legend for a better understanding of the figure.

For Figure 3, c, you can include the Faiths phylogenetic diversity plot here as 3A. You can rename the x-axis as the concentration and not letter codes. The current format is somehow difficult to interpret or read. It’s perhaps more intuitive to show the barplot of all replicates (not pooled) so we can see the relative distribution of all the samples in each treatment. It is also better to plot each sample, so we know how many samples were sequenced. You can put 3B in supplements since this was not frequently mentioned in the results and discussion.

Incorporate p-values for the variance of PCA in Figure 4.

Consider replacing Figure 5 with a figure displaying the enriched GO terms (e.g., BP of GOMWU) in the treatment groups. This way, you could highlight the specific processes discussed in the manuscript.

Reviewer 3 ·

Basic reporting

please see attached document

Experimental design

please see attached document

Validity of the findings

please see attached document

Additional comments

please see attached document

Annotated reviews are not available for download in order to protect the identity of reviewers who chose to remain anonymous.

---

## Round 0.2 · Minor Revisions

A few minor issues remain.

Reviewer 1 ·

Basic reporting

Thank you for the new version of the manuscript. However, the following points were not answered satisfactorily:

Experimental design

Line 306: With this experimental setup, direct or indirect effects of antibiotics on host gene expression cannot be separated. Indirect effects might be mediated by the changed bacterial communities. To separate these effects, germfree animals need to be analyzed with and without antibiotics. This point needs to be discussed. For bacterial effects on Nematostella gene expression see https://www.biorxiv.org/content/10.1101/2022.12.13.520252v1.

Response authors: The antibiotic acute condition is the germfree animals that have been removed from the antibiotics. The antibiotic constant condition is germfree animals that have remained in antibiotics. Moreover, this point was discussed later in the discussion which was addressed by other comments.

Response reviewer: It is not clear from the results that the acute antibiotic acute condition did not lead to bacterial growth. Since bacteria still grow after most antibiotic treatments, these animals were probably not germ-free. Therefore, the argument is not convincing. If the antibiotic acute condition is really germfree, please separate direct and indirect antibiotic effects in the analyses.

Validity of the findings

Line 171: What is the reason for using the threshold of OD>1 for detecting bacterial growth. Many bacteria do not reach an OD of 1.0 in liquid media. Did the authors include a negative control (media with no turbidity) in the subsequent 16S rRNA sequencing?

Response authors: When running these experiments, the OD had a binary outcome, either remained at control OD600 ~ 0.1, or growth reached or exceeded 1. We used OD600 = 1 as an arbitrary value, and our results did not indicate that we should have reduced this value.

We did not include a negative control in sequencing due to the binary nature of the turbidity experiments.

Response reviewer: From the microbiology point of view this is not convining. An OD600 from 0.1 indicates clear bacterial growth. What was the OD600 from negative controls? That should be 0.0.



Line 283: It is not clear to me what conclusions can be drawn from the data of the cultured bacteria. The growth of bacteria in liquid culture leads to an over presence of fast-growing bacteria. Therefore, these data allow only a small conclusion about the bacterial community of the host.

Response authors: We are concluding that there is growth following removal of the antibiotic stressor. While the proportions may not be an accurate representation of the bacteria that remain in the animal following antibiotic exposure, the authors are confident that we have encapsulated the ASVs that have the potential to recolonize the host.

Response reviewer: The introduced bias needs to be discussed critically.


Line 169: The authors should clarify the reasons for using liquid cultures vs. agar plates? During growth in liquid media fast growing bacteria overgrow slow growing bacteria and get overestimated.

Response authors: The authors used liquid media because it better represented the environment the bacteria would experience in the environment.

Response reviewer: This argumentation is not convincing and does not address the criticism of the biased approach of culturing in liquid media.

Reviewer 3 ·

Basic reporting

The authors have taken care to address all comments raised by the reviewers and heavily revised all sections of the manuscript. The aim of the study and the conclusions are now clearer. Also some methodological draw-backs are now addressed in the discussion.

Experimental design

no additional comments

Validity of the findings

no additional commentsJust some minor notes

Additional comments

Just some minor notes

l.24 and 103, 357, 484: Alpha- and Gammaproteobacteria capital letters
l.69 critical not only for
l.505 the definition of resistance is that they are not affected by an antibiotic at any concentration; so it might suggest that they are only susceptible at high concentraitons or less susceptible or more resilient
l.509 maybe put into context with literature references where rifampicin and combinations had a stronger effect as well. Certainly combinations target more different bacteria than a single antibiotic
l.585 impact the immune system

---

## Round 0.3 · accepted · Accept

Thank you for replying to the last comments, your paper can now be accepted for publication.